# Rewarding the Journey, Not Just the Destination: A Composite Path and Answer Self-Scoring Reward Mechanism for Test-Time Reinforcement Learning

## Abstract

Most recently, Reinforcement Learning (RL) has empowered frontier Large Language Models (LLMs) to solve challenging math, science, and coding problems. This paper consentrates on RL on data without explicit labels for reasoning tasks in LLMs. The core challenge of the problem is reward estimation during inference in absense of ground-truth information. In this work, we propose **COMPASS**: **Com**posite **P**ath and **A**nswer **S**elf-**S**coring - a novel method for training LLMs using RL on unlabeled test data. **COMPASS** consists of Dual-Calibration Answer Reward (DCAR) and Decisive Path Reward (DPR), which enables self-evolution of LLMs by fully utilizing the priors in the pre-trained models as intrinsic rewards. We find that by simultaneously reinforcing the trustworthy consensus answers and chains of thought that yield high model desiciveness on its generated responses, the model improves its reasoning ability. Our experiments demonstrate that COMPASS consistently improves performance across a variety of tasks and models, marking a further step of learning from continuous streams of experience.

## 1 Introduction

Reinforcement Learning (RL) (Kaelbling et al., 1996; Dong et al., 2024) has proven to be an effective pathway for pushing the capability ceiling of pre-trained Large Language Models (LLMs) in complex tasks such as mathematical reasoningSetlur et al. (2024); Gao et al. (2024); Albright & Andemicael (2025) and code generation (Guo et al., 2025; Achiam et al., 2023; Team, 2025; Wang et al., 2024; Islam et al., 2024). However, most existing methods rely heavily on external supervision (Luong et al., 2024), rewarding models (Ouyang et al., 2022; Shao et al., 2024) based on correctness with respect to ground-truth labels, which significantly limits their scalability. As real-world tasks continue to increase in both complexity and volume, large-scale annotation for RL becomes increasingly impractical, posing a substantial barrier to the continual improvement of state-of-the-art models.

Further building upon the substantial progress of LLMs, it naturally motivates a promising direction in which AI systems autonomously improve via RL on unlabeled data by directly engaging in self-experience and learning, thereby pushing the boundaries of RL and further advancing the frontier of AI capabilities(Zhang et al., 2025c; Xiong et al., 2025; Huang et al., 2024). Against this backdrop, TTRL (Test-Time Reinforcement Learning) (Zuo et al., 2025) was the first to propose the task of updating model parameters during test time based on unlabeled test data. This novel setting has recently attracted increasing attention. This work also focuses on the adaptation to test-time data.

Therefore, we aim to fully advance AI evolution by updating models at test time using RL, thereby enhancing their generalization to previously unseen data. However, this introduces a critical challenge: ***How to obtain rewards for RL through LLMs' own experience?*** TTRL (Zuo et al., 2025) achieves this by repeatedly sampling multiple responses for the same problem, then constructing pseudo-labels (and rewards) based on the self-consistency consensus derived from majority voting. The essence of this approach is to rely on the model's intrinsic confidence as a proxy metric when external rewards are unavailable. In other words, an answer consistently reproduced by the model

across multiple trials is considered higher-confidence in correctness. This is analogous to human problem-solving, where a conclusion verified through diverse methods strengthen our confidence in its correctness. The success of TTRL validates the effectiveness of optimizing internal confidence proxies to indirectly enhance reasoning capabilities.

However, while TTRL represents a significant step towards autonomous learning, we argue that its current self-rewarding mechanism, based on majority voting, suffers from two fundamental limitations. ***First is the fragility of pseudo-labels and the sparsity of reward signals.*** An erroneous consensus can mislead the model, while the binary reward signal is too sparse to distinguish high-quality consensus from fortuitous agreement, thereby limiting the efficiency and stability of the learning process. ***Second is the neglect of reasoning process quality.*** TTRL is a purely outcome-based method; it rewards the final answer for matching the pseudo-label but fails to evaluate the quality of reasoning chains themselves. This creates the risk of the model reinforcing a flawed reasoning process that coincidentally yields the correct answer, fundamentally capping its potential to improve its logical reasoning capabilities.

To address these challenges, we propose **Com**posite **P**ath and **A**nswer **S**elf-**S**coring (**COMPASS**) reward mechanism. COMPASS consists of Dual-Calibration Answer Reward (DCAR) and Decisive Path Reward (DPR). DCAR calibrates majority voting with confidence measures and further evaluates the credibility of pseudo-labels, which enable the model to prioritize learning from high-credibility consensus, effectively enhancing learning stability and efficiency. Furthermore, we introduce Decisive Path Reward (DPR), which moves beyond the final answer to scrutinize each step of the generation process. Through an entropy-weighting mechanism, it encourages the model to make more decisive choices (high decisiveness) at critical junctures of high uncertainty (high entropy), providing a direct and dense supervisory signal for optimizing the reasoning path.

The main contributions can be summarized as follows: (1) We propose **COMPASS**, a novel self-scoring reward mechanism for reinforcement learning on unlabeled data. COMPASS is designed to enable the self-evolution of LLMs by generating intrinsic rewards that evaluate both the final answer and the intermediate reasoning path, addressing the key limitations of outcome-only reward signals in prior work. (2) We design a composite reward function consisting of two novel components: Dual-Calibration Answer Reward (DCAR) and Decisive Path Reward (DPR). DCAR moves beyond simple majority voting by dual-calibrating consensus, yielding more reliable reward for the final answer. DPR introduces a process-centric evaluation that provides dense rewards for decisive token generation during uncertain steps in the reasoning chain, directly optimizing the quality of the thought process. (3) Extensive experiments on diverse reasoning benchmarks demonstrate the effectiveness and superiority of the proposed COMPASS.

## 2 RELATED WORK

### 2.1 RL FOR REASONING

Table 1: Evolution of RL Paradigms from External Supervision to Internal Feedback

| Paradigm | Reward Source | Requirements | Task |
|----------|---------------|--------------|------|
| **RLHF** | Learnable Model | Human Preference Data | General Questions |
| **RLVR** | Rule-based Function | Gold-Standard Answers | Math/Code Questions |
| **RLIF** | Rule-based Function | None (Self-supervised) | Math/Code Questions |

Reinforcement Learning (RL) plays a critical role in enhancing the instruction-following capabilities of Large Language Models (LLMs) (Guo et al., 2025; Ouyang et al., 2022). Its evolution can be broadly categorized into three successive paradigms: Reinforcement Learning from Human Feedback (RLHF) (Ouyang et al., 2022; Zheng et al., 2023; Dai et al., 2023), Reinforcement Learning with Verifiable Rewards (RLVR) (Luong et al., 2024; Shao et al., 2024), and Reinforcement Learning from Internal Feedback (RLIF) (Zhang et al., 2025a; Zuo et al., 2025; Zhao et al., 2025), each gradually reducing the dependency on external supervision.

RLHF aligns base models with human preferences using algorithms such as Proximal Policy Optimization (PPO) (Schulman et al., 2017), where the human-annotated preference data is essential. Recently, Large Reasoning Models (LRMs), such as DeepSeek-R1 (Guo et al., 2025), have demonstrated the significance of RL in improving reasoning abilities using verifiable rewards. These RLVR methods reduce reliance on nuanced human feedback but still operate on labeled training datasets (i.e., they require ground-truth answers to generate rewards). Against this backdrop, the RLIF paradigm (or self-rewarding) has emerged, aiming to learn from unlabeled data. For instance, EMPO (Zhang et al., 2025a) incentivizes reasoning capabilities by minimizing the entropy of LLM generations in a latent semantic space in a fully unsupervised manner. Similarly, TTRL (Zuo et al., 2025) employs repeated sampling strategies to generate pseudo-labels via consensus, which are then used to compute rule-based rewards, thereby facilitating efficient RL without ground-truth labels.

## 2.2 CONFIDENCE-BASED REWARD

In fully unsupervised settings (Zhang et al., 2025b; Wei et al., 2025; Basavatia et al., 2024) where ground-truth labels are unavailable, directly optimizing for the correctness of Large Language Model's (LLM) outputs is infeasible. Consequently, a promising direction is to identify an intrinsic proxy metric highly correlated with correctness. Model confidence (Xiong et al., 2023; Tripathi et al., 2025; Tian et al., 2025) has emerged as a natural and effective candidate for this role. This approach mirrors human cognition, where confidence often serves as the primary internal signals for guiding reasoning when not having access to external feedback information.

Language models output distributions over tokens, and the confidence is typically quantified based on these distributions. Specifically, confidence can be measured by the log-likelihood of generated sequences, where higher values signal greater model conviction, or conversely, by the entropy of these distributions, where lower entropy signifies higher certainty. Another alternative paradigm assesses confidence through self-consistency, where multiple responses are sampled for the same prompt. Specifically, RLSC (Li et al., 2025) reinforces responses by maximizing log-likelihood. RENT (Prabhudesai et al.) minimizes entropy to promote high-certainty outputs. And TTRL (Zuo et al., 2025) estimates pseudo-labels via self-consistency with majority voting, further reinforcing consensuses. Experimental results show that these methods have all led to better reasoning performance.

## 2.3 TEST-TIME ADAPTATION

Test-Time Adaptation (TTA) (Sun et al., 2017; Maria Carlucci et al., 2017; Schneider et al., 2020) is where a model is updated using unlabled test data during test time. The goal is to improve performance in scenarios where there is a distribution shift between training and testing environments. Tent (Wang et al., 2020) performs entropy minimization on model predictions during test time, which assumes that predictions on test data should be low in entropy if the model is well-adapted to the new distribution. And TTRL (Zuo et al., 2025) proposed test-time reinforcement learning using majority voting as a reward.

## 3 COMPOSITE PATH AND ANSWER SELF-SCORING (COMPASS)

This section introduces our proposed Composite Path and Answer Self-Scoring (COMPASS) reward mechanism. Building on TTRL (Zuo et al., 2025), we identify and address several limitations in its reward mechanism. Our work enhances the robustness and efficiency of test-time adaptation by introducing a refined reward system that incorporates both outcome and process rewards.

### 3.1 MOTIVATION: LIMITATIONS OF TTRL

TTRL (Zuo et al., 2025) operates on the principle of self-consistency with majority voting. In this process, a pseudo-label is generated by majority voting among multiple candidate outputs, which is then used to form a binary reward signal for optimizing LLMs. However, we identify two fundamental limitations:

1. **Fragility of Pseudo-Labels**: The effectiveness of TTRL is highly dependent on the quality of pseudo-labels derived from majority voting. In cases of incorrect consensus, the model

is driven towards suboptimal policies. Additionally, the binary reward signal is sparse and fails to reflect the reliability of pseudo-labels, limiting the efficiency of the learning process.

2. **Neglect of Reasoning Quality**: TTRL is an outcome-based method, focusing solely on the final prediction rather than the reasoning steps leading to the output. Although a correct answer can result from a flawed reasoning process—and an incorrect one from a minor error in a sound chain of thought—TTRL lacks direct supervision of the reasoning steps, which limits its capacity to improve LLMs' reasoning abilities.

To overcome these challenges, we propose a novel Composite Path and Answer Self-Scoring (COM-PASS) reward mechanism, which combines the outcome-based Dual-Calibration Answer Reward (DCAR) and process-based Decisive Path Reward (DPR) to provide more informative learning signals for RL, accounting for both the final output and the underlying decision-making process.

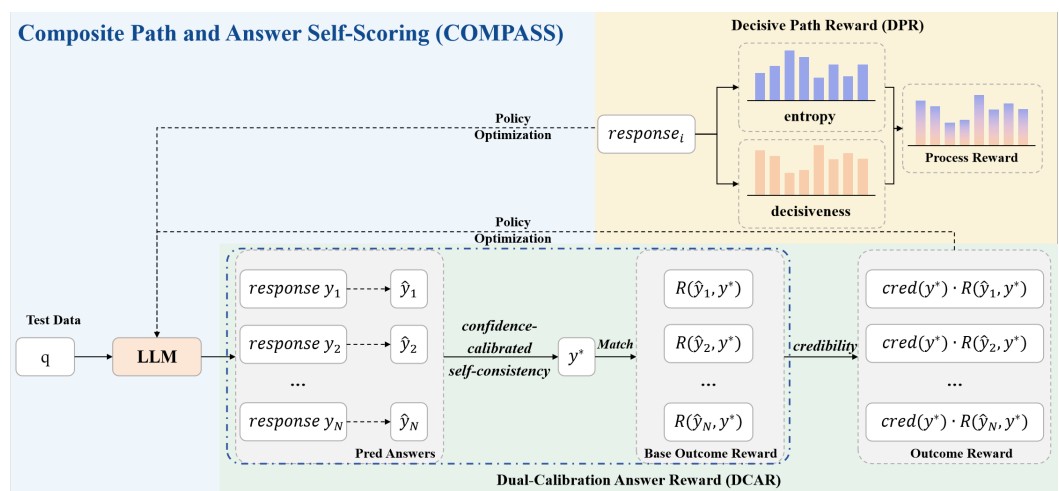

Figure 1: The Composite Path and Answer Self-Scoring (COMPASS) reward mechanism.

## 3.2 OVERALL FRAMEWORK

Figure 1 illustrates how our approach, COMPASS, achieves test-time reinforcement learning. Given a state represented by the prompt $q$, the model acts by producing an output $y$ sampled from a policy $\pi_\theta (y \mid q)$ parameterized by $\theta$. To construct reward signals without ground-truth labels, we generate multiple candidate responses and extract the corresponding answers $\{\hat{y}_1, \hat{y}_2, \ldots, \hat{y}_N\}$ from the model through repeated sampling. A consensus output $y^*$ is derived through ***confidence-calibrated self-consistency***, serving as a proxy for the optimal action. The environment then provides a reward $R(\hat{y}_i, y^*)$ based on the alignment between the sampled action $\hat{y}_i$ and the consensus action $y^*$. To further stabilize RL training, we propose a ***credibility*** metric to assess the quality of generated pseudo-labels. In addition to the aforementioned outcome-based Dual-Calibration Answer Reward (DCAR), we also introduce the process-based Decisive Path Reward (DPR) to evaluate the reasoning quality of each candidate response. The total reward is a combination of the two components:

$$R(y_i) = R_{\text{outcome}}(y_i) + R_{\text{process}}(y_i) \tag{1}$$

where $R_{\text{outcome}}(y_i)$ represents the DCAR outcome reward, and $R_{\text{process}}(y_i)$ is the DPR process reward. The composite reward $R(y_i)$ is then used to update the model parameters $\theta$ via policy gradient methods.

## 3.3 OUTCOME REWARD: DUAL-CALIBRATION ANSWER REWARD (DCAR)

To address the fragility of pseudo-labels from majority voting, we propose a dual calibration method. First, we calibrate the self-consistency mechanism using a confidence metric to generate a more reliable pseudo-label and its corresponding base binary reward. Subsequently, we introduce a credibility score to modulate this base reward, thereby implementing a form of soft curriculum learning. The complete calculation process of DCAR refers to algorithm 1.

### 3.3.1 CONFIDENCE-CALIBRATED SELF-CONSISTENCY

In the standard self-consistency mechanism, each candidate's vote is treated equally. ***We hypothesize that more confident responses should contribute more significantly to the final decision.*** Through empirical analysis, we found that the standard deviation of ***topk_diff*** (the probability difference between the top-1 and top-2 tokens on each token) across a generation trajectory correlates strongly with the correctness of the final answer, which captures the model's stability and decisiveness in prediction. Thus, we define the ***confidence*** of a trajectory $\hat{y}_i$ as:

$$\text{topk\_diff}(x_t) = p(x_t^1 | x_{<t}) - p(x_t^2 | x_{<t}) \tag{2}$$

$$\text{confidence}(\hat{y}_i) = \exp\left(-\text{std}_t \text{ topk\_diff}(x_t)\right) \tag{3}$$

where $x_t^1$ and $x_t^2$ represent the top-1 and top-2 tokens at timestep $t$. And we define ***confidence*** as the negative exponential of the standard deviation (std), which ensures lower std yields higher confidence and guarantees positive weights required for self-consistency. Then we propose ***confidence-calibrated self-consistency***. The score for a given answer $y$ is the confidence-weighted sum:

$$\text{confidence-calibrated-self-consistency}(y) = \frac{\sum_{i:\hat{y}_i=y} \text{confidence}(\hat{y}_i)}{\sum_{i=1}^{N} \text{confidence}(\hat{y}_i)} \tag{4}$$

The final pseudo-label $y^*$ is determined as the answer with the highest calibrated score:

$$y^* = \arg\max_y \text{confidence-calibrated-self-consistency}(y) \tag{5}$$

### 3.3.2 CREDIBILITY-CALIBRATED PSEUDO-LABELS

Once the confidence-calibrated pseudo-label $y^*$ is obtained, it is essential to assess its credibility. ***Our underlying hypothesis is that a consensus derived from high-confidence responses is more reliable than one based on diverse low-confidence outputs.*** Therefore, we propose a ***credibility*** metric based on two key concepts:

(1) General Group: The General Group contains all responses that agree with the pseudo-label $y^*$. The group's confidence $C_{\text{General}}$ is defined as the maximum confidence within this subset, representing the strongest supporting evidence for the consensus:

$$C_{\text{General}} = \max_{i:\hat{y}_i=y^*} C(\hat{y}_i) \tag{6}$$

(2) Elite Response: This is the response among all $N$ candidates with the highest confidence, denoted as $C_{\text{Elite}}$:

$$C_{\text{Elite}} = \max_{i=1,\ldots,N} C(\hat{y}_i) \tag{7}$$

The ***credibility*** of the pseudo-label $y^*$ is the ratio of these two confidence scores:

$$\text{credibility}(y^*) = \frac{C_{\text{General}}}{C_{\text{Elite}}} \tag{8}$$

This ratio quantifies the strength of the consensus relative to the most confident individual opinion. A value of 1 indicates that the most confident response aligns with the consensus, suggesting high reliability, while a value less than 1 indicates the presence of a highly confident dissenter, thereby lowering trust in the consensus.

---

**Algorithm 1:** Outcome Reward: DCAR

**Input:** Prompt $q$, policy $\pi_\theta$, number of samples $N$.
**Output:** outcome rewards $\{R_{\text{outcome}}(y_i)\}_{i=1}^N$.
**Initialize:**
    trajectories $Y \leftarrow [\,]$,
    answers $\hat{Y} \leftarrow [\,]$,
    confidences $C \leftarrow [\,]$;
`// S1:Sample & Calc Confidence`
**for** $i \leftarrow 1$ **to** $N$ **do**
    Sample trajectory $y_i \sim \pi_\theta(\cdot|q)$;
    Extract final answer $\hat{y}_i$ from $y_i$;
    $c_i = \exp(-\text{std}_t \text{ topk\_diff}(x_t))$; `// Eq.(3)`
    Append to $Y, \hat{Y}, C$;

`// S2:Get Pseudo-Label`
Find the set of unique answers $A = \text{unique}(\hat{Y})$;
Initialize answer scores: $S[a] \leftarrow 0$ for all $a \in A$;
**for** $i \leftarrow 1$ **to** $N$ **do**
    $S[\hat{y}_i] \leftarrow S[\hat{y}_i] + c_i$;
$y^* = \arg\max_{a \in A} S[a]$;      `// Eq.(5)`
`// S3:Calc Credibility`
$C_{\text{General}} \leftarrow \max(\{c_i \mid \hat{y}_i = y^*\})$;    `// Eq.(6)`
$C_{\text{Elite}} \leftarrow \max(\{c_i\})$;           `// Eq.(7)`
$\text{cred} \leftarrow C_{\text{General}}/C_{\text{Elite}}$;      `// Eq.(8)`
`// S4:Compute Reward`
**for** $i \leftarrow 1$ **to** $N$ **do**
    **if** $\hat{y}_i = y^*$ **then**
        $R_{\text{base}} \leftarrow 1$;
    **else**
        $R_{\text{base}} \leftarrow 0$;
    $R_i \leftarrow \text{cred} \cdot R_{\text{base}}$;        `// Eq.(9)`
    Append $R_i$ to $R_{\text{outcome}}$;
**return** $\{R_i\}_{i=1}^N$;

### 3.3.3 DUAL-CALIBRATION ANSWER REWARD

The final outcome reward is obtained by combining the above components. The base reward $R_{\text{base}}(\hat{y}_i)$ provides a binary signal indicating whether $\hat{y}_i$ matches the pseudo-label $y^*$, which is then modulated by the credibility score:

$$R_{\text{outcome}}(\hat{y}_i) = \text{credibility}(y^*) \cdot R_{\text{base}}(\hat{y}_i)$$
$$= \text{credibility}(y^*) \cdot \begin{cases} 1, & \text{if } \hat{y}_i = y^* \\ 0, & \text{otherwise} \end{cases} \qquad (9)$$

This weighting operation transforms the sparse binary reward into a continuous signal in the range $[0, 1]$, effectively implementing a soft curriculum learning mechanism. The model is encouraged to focus on pseudo-labels with high credibility, promoting more stable and accurate learning.

### 3.4 PROCESS REWARD: DECISIVE PATH REWARD (DPR)

DCAR provides a robust signal for the quality of the final pseudo-label. To move beyond outcome-based supervision, we also introduce the Decisive Path Reward (DPR). DPR is specifically designed to scrutinize the model's reasoning pathway, encouraging decisive actions at critical junctures to ensure the integrity of the entire reasoning chain. Specifically, we evaluate two metrics at each token generation step $t$:

- **Decisiveness**: Quantified by ***topk_diff***. A larger value signifies higher decisiveness, indicating a more confident and unambiguous decision point.

$$\text{decisiveness}(x_t) = \text{topk\_diff}(x_t) \qquad (10)$$

- **Uncertainty**: Quantified by ***entropy*** ($H_t$) of the model's predictive distribution. A higher value signifies greater uncertainty, indicating a more challenging decision point.

***Our central hypothesis is that decisiveness is more valuable during moments of high uncertainty.*** A confident action is more informative when the model faces multiple viable alternatives. Based on this premise, we define the process reward, $R_{\text{process}}(y_i)$, by dynamically weighting the decisiveness of each step in the trajectory by its corresponding uncertainty (entropy):

$$w_t = \frac{e^{H_t}}{\sum_{j=1}^{T} e^{H_j}} \qquad (11)$$

$$R_{\text{process}}(\hat{y}_i) = \sum_{t=1}^{T} w_t \cdot \text{decisiveness}(x_t) \qquad (12)$$

By providing a dense, per-token feedback signal, the process reward incentivizes the model to execute decisive actions at critical, high-uncertainty junctures, thereby fostering the development of more robust reasoning paths.

## 4 EXPERIMENTS

### 4.1 EXPERIMENTAL SETUP

**Models** To evaluate the generalizability of COMPASS across different backbone models, we conduct experiments using both base and instruct models of various scales. The models we experiment with include the instruct model LLaMA-3.2-1B-Instruct (Meta, 2024), the math base model Qwen2.5-Math-1.5B (Yang et al., 2025), and the vanilla base model Qwen2.5-7B (Yang et al., 2025).
**Benchmarks** We evaluate COMPASS on GPQA-Diamond (Rein et al., 2024), a challenging and high-quality subset of the Graduate-Level Google-Proof Question Answering benchmark, and 3 mathematical reasoning benchmarks: AIME 2024 (Yuan et al., 2024), AMC (Yuan et al., 2024), and MATH-500 (Hendrycks et al., 2021).

**Evaluation Setup** We apply COMPASS to each benchmark individually and then evaluate. For the main experiments, following DeepSeek-R1 (Guo et al., 2025), we adopt the pass@k evaluation

protocol and report pass@1 using non-zero temperature sampling. Specifically,we generate 16 responses (4 for 32k context) per question using a temperature of 0.6 and a top-$p$ value of 0.95. The pass@1 score is computed as:

$$\text{pass@1} \ = \frac{1}{k}\sum_{i=1}^{k} p_i \tag{13}$$

where $p_i$ indicates whether the $i$-th response is correct.

**Baselines** Since TTRL (Zuo et al., 2025) firstly proposed the test-time reinforcement learning task, we primarily compare our COMPASS with the backbone model and TTRL baselines to validate whether COMPASS can achieve effective improvements through self-evolution.

**Implementation Details** We independently apply GRPO (Shao et al., 2024) on each benchmark to implement COMPASS. For hyperparameters, we use a cosine learning rate schedule with a peak value of $5 \times 10^{-7}$ and adopt the AdamW optimizer for the policy model. For rollout, we sample 64 responses using a temperature of 0.6 (1.0 for Qwen2.5-Math (Yang et al., 2025)) for voting-based label estimation and downsample 32 responses per prompt for training. Evidence shows that our Composite Path and Answer Self-Scoring reward strategies retain relatively low computational costs while still achieving strong performance. For models with fewer than 7B parameters, we follow the settings of the original TTRL paper. We set the number of episodes to 10, 30, and 80 for MATH-500, AMC, and AIME 2024, respectively, based on the dataset size. For the larger Qwen2.5-7B model, we reduced the number of training epochs to investigate the methods' performance under a more computationally efficient regime, which ensures a fair comparison between the methods under the same resource constraints. The corresponding number of epochs for these datasets were 2, 8, and 20, respectively (approximately 20% of those in the original TTRL paper).

## 4.2 MAIN RESULTS

Table 2: Performance comparison on test-time reinforcement learning.

| Name | AIME | AMC | MATH | GPQA |
|------|------|-----|------|------|
| **Instruct Models** | | | | |
| *LLaMA3.2-1B-Instruct* | 1.5 | 9.8 | 24.7 | 23.8 |
| *TTRL* | 6.7 | 19.2 | 27.8 | 24.0 |
| ***COMPASS*** | 3.5 | 20.1 | 28.7 | 25.8 |
| $\Delta$ | -3.1 | +0.9 | +0.9 | +1.8 |
| **Math Base Models** | | | | |
| *Qwen2.5-Math-1.5B* | 7.7 | 28.6 | 32.7 | 24.9 |
| *TTRL* | 15.8 | 47.4* | 72.4* | 26.1 |
| ***COMPASS*** | 18.3 | 48.6 | 73.1 | 29.3 |
| $\Delta$ | +2.5 | +1.2 | +0.7 | +3.2 |
| **Vanilla Base Models** | | | | |
| *Qwen2.5-7B[†]* | 7.5 | 34.6 | 60.9 | 30.5 |
| *TTRL* | 20.0 | 50.2 | 76.6 | 31.1 |
| ***COMPASS*** | 23.5 | 53.2 | 76.9 | 31.7 |
| $\Delta$ | +3.5 | +3.0 | +0.3 | +0.6 |

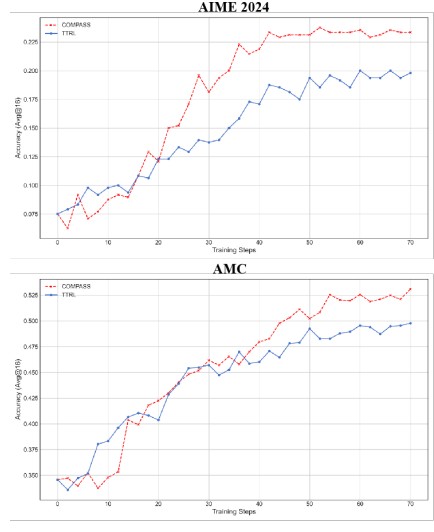

Figure 2: Performance comparison on AIME/AMC using Qwen2.5-7B.

***COMPASS performs well on most tasks and models.*** Table 2 presents the main results. We apply COMPASS to 3 models spanning 3 model families, 2 model types, and 3 model sizes, consistently demonstrating obvious improvements across 4 highly challenging benchmarks. On the AIME 2024 and GPQA benchmarks, COMPASS achieves significant improvements of 15.8% and 12.3%, respectively, over TTRL when using Qwen2.5-Math-1.5B. For experiments with the larger Qwen2.5-7B base model, both TTRL and COMPASS were trained for approximately 20% of the epochs specified in the original TTRL paper due to computational constraints. Despite this reduced training schedule, our method demonstrates a clear and consistent performance advantage over TTRL, as evidenced by both the final evaluation metrics in Table 2 and the performance trend curves illustrated

in Figure 2. However, we note an exception with the LLaMA3.2-1B-Instruct model on the AIME 2024 dataset. We attribute this performance drop to the model's insufficient foundational knowledge. For such a model, the high-entropy states targeted by our process reward (DPR) likely signify fundamental confusion rather than meaningful reasoning junctures. Reinforcing these spurious signals inadvertently degrades performance, highlighting that the efficacy of our method relies on the base model possessing a solid knowledge foundation.

***COMPASS naturally scales.*** As shown in Table 2, another noteworthy observation is that as the model size increases (1B →1.5B → 7B), performance consistently improves, highlighting the natural scaling behavior of COMPASS: larger models can produce more accurate rewards during self-improvement, which leads to more effective learning on new data. ***COMPASS achieves sustainable***

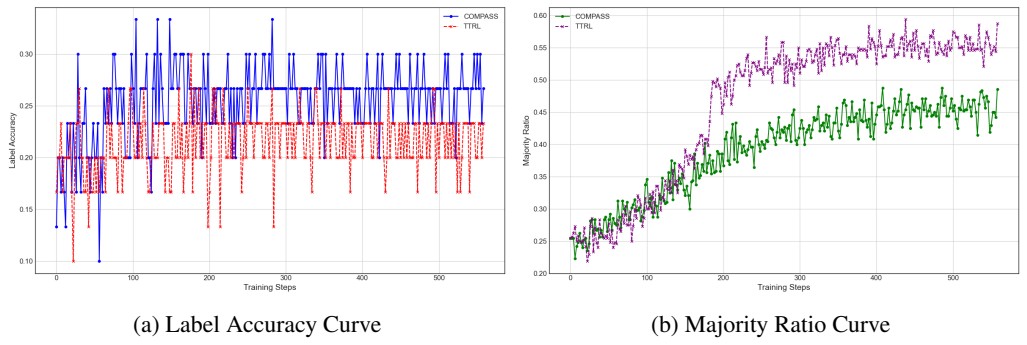

| (a) Label Accuracy Curve | (b) Majority Ratio Curve |

Figure 3: Training dynamics comparision on AIME using Qwen2.5-Math-1.5B.

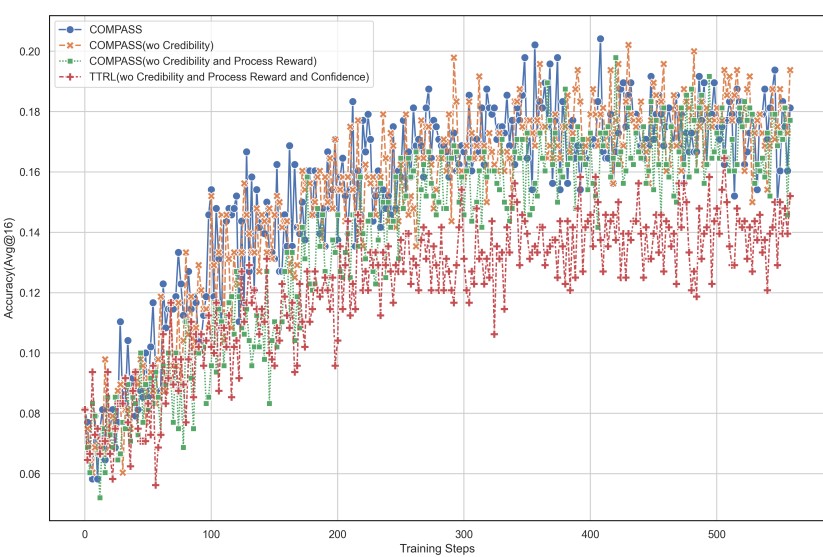

Figure 4: Ablation Results of COMPASS on AIME using Qwen2.5-Math-1.5B.

***self-evolution through online and RL.*** To understand the mechanisms of our proposed COMPASS framework, we analyzed its training dynamics against the TTRL baseline as shown in Figure 3, focusing on ***pseudo-label accuracy*** and ***majority ratio***. The results highlight COMPASS's superior learning process. Our method achieves significantly higher and steadily improving pseudo-label accuracy, confirming that its advanced reward system—which combines the outcome-based Dual-Calibration Answer Reward (DCAR) and process-based Decisive Path Reward (DPR), and generates more effective training signals. In contrast, the baseline's accuracy stagnates at a much lower level. Simultaneously, COMPASS maintains a consistently lower majority ratio. This demonstrates that it successfully avoids the baseline's tendency to prematurely converge on the most frequent answer, a common pitfall of naive majority voting. Instead of simply reinforcing the consensus, COMPASS

values diverse and high-quality reasoning paths. ***This dual dynamics of increasing label accuracy while reducing reliance on a single popular answer provides strong evidence for COMPASS's effectiveness.*** It cultivates a more robust self-evolution by considering both the intrinsic quality of reasoning paths and the popularity of final answers, leading to a more reliable self-improving model.

### 4.3 ABLATION RESULTS

COMPASS is built upon TTRL and incorporates both outcome reward and process reward. The outcome reward, named Dual-Calibration Answer Reward (DCAR), involves dual calibration through confidence and credibility assessments, while the process reward corresponds to the Decisive Path Reward (DPR). We conducted sequential ablative experiments by progressively removing components: eliminating credibility calibration, then process reward, and finally confidence calibration, ultimately reverting to the baseline TTRL. The performance curves of these models across training steps (as shown in Figure 4) demonstrate that all three components can enhance model performance. Notably, confidence calibration contributes the most significant performance improvement, evidenced by the largest vertical distance between the red and green curves.

### 4.4 CASE STUDY

To illustrate the superiority of COMPASS, we present a case study where the model's responses lead to a competing consensus (Figure 5) —— the incorrect answer '32' and the correct answer '116' both received two votes each, creating a tie. The TTRL baseline using majority voting is unable to resolve this tie and arbitrarily selects the incorrect label '32'. In contrast, COMPASS weighs each vote by its confidence score. As shown in Figure 5b, the cumulative confidence for the correct answer '116' (**1.6235**) is slightly higher than for '32' (1.6175). This crucial, fine-grained signal allows COMPASS to break the tie correctly and select '116' as the pseudo-label. This case clearly demonstrates our method's robustness in scenarios where simple voting mechanisms fail.

(a) A curated summary of key model responses.

| Resp. ID | Answer | Correct? | Confidence | Reasoning Path Snippet |
|----------|--------|----------|------------|------------------------|
| #2 | 32 | ✗ | 0.8327 | `To solve the problem...` |
| #7 | 32 | ✗ | 0.7848 | `To solve the problem...` |
| #12 | 116 | ✓ | 0.8262 | `To solve the problem...` |
| #15 | 116 | ✓ | 0.7973 | `Let's break down...` |

(b) Comparison of the final decision by each voting mechanism.

| Mechanism | Decision Basis | Pseudo-Label | Result |
|-----------|----------------|--------------|--------|
| TTRL | Majority voting (2 vs. 2 tie) | `"32"` | ✗ Failed |
| **COMPASS** | Confidence-calibrated score (**1.6235** vs. 1.6175) | `"116"` | ✓ Succeeded |

Figure 5: A Case Study on AIME 2024 (Ground Truth: 116).

## 5 CONCLUSION

In this paper, we propose a novel Composite Path and Answer Self-Scoring (COMPASS) reward mechanism for training Large Language Models with Reinforcement Learning on test data without access to ground-truth labels. It combines the outcome-based Dual-Calibration Answer Reward (DCAR) and process-based Decisive Path Reward (DPR) to collaboratively address two core limitations in the prevailing self-rewarding mechanism: the fragility of pseudo-labels and the disregard for the reasoning process. Our experiments demonstrate the strong potential of COMPASS, achieving consistent improvements across a variety of tasks. We view COMPASS as a further step toward RL with self-labeled rewards, marking an important direction of learning from continuous streams of experience.

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

# A APPENDIX

## A.1 LARGE LANGUAGE MODELS USAGE STATEMENT

In the preparation of this research, large language models (LLMs) were employed strictly as a limited-purpose auxiliary tool. The models were used exclusively for language polishing tasks, including grammar checking, sentence structure optimization, and wording refinement to improve the readability and linguistic fluency of portions of the text. The LLMs played no role in any core research activities, including but not limited to: research ideation, theoretical development, experimental design, data analysis, result interpretation, or scientific decision-making. All intellectual contributions to this work originate solely from the human authors. The authors take full responsibility for the entire content of this paper, including text polished by LLMs, and affirm its originality, accuracy, and academic integrity.

