# OpenReview forum: "Rewarding the Journey, Not Just the Destination: A Composite Path and Answer Self-Scoring Reward Mechanism for Test-Time Reinforcement Learning"
_ICLR.cc/2026/Conference — ICLR 2026 Conference Withdrawn Submission_

### Official Review · Reviewer_NWaM · 2025-10-25

**Soundness:** 3
**Presentation:** 3
**Contribution:** 2
**Rating:** 4
**Confidence:** 3

**Summary:**

Based on TTRL, this paper further explores the direction of RLIF and proposes the COMPASS method. To address the reward design for reasoning tasks without labels, COMPASS introduces solutions from two dimensions: **outcome** and **process**.

1. **Outcome Reward**: It adopts **Confidence-Calibrated Self-Consistency**, where the *topkdiff* metric is used to characterize the model's confidence level. Additionally, **Credibility-Calibrated Pseudo-Labels** are applied to eliminate abnormal cases.

2. **Process Reward**: It employs the **Decisive-Path-Reward** to represent the value degree of each generated token.

Furthermore, small-scale experiments are conducted to verify that COMPASS outperforms the TTRL method, and additional ablation studies are also carried out to validate the effectiveness of each component in COMPASS.

**Strengths:**

1. The paper is composed in a clear and fluent style, making it easy to read and understand.
2. The proposed method is simple and clearly defined, facilitating comprehension.
3. The approach to designing rewards from both **outcome** and **process** perspectives shows a degree of innovation.
4. The experimental results indicate that the method achieves observable performance improvements compared to TTRL.

**Weaknesses:**

1. The paper presents numerous **hypotheses** throughout, yet lacks experimental validation or theoretical derivation to substantiate their validity;

2. The connection between **Decisive-Path-Reward** and **process reward** appears somewhat tenuous—assigning rewards to tokens does not inherently qualify as a process reward mechanism;

3. The experimental scope is limited to models up to **7B parameters**, which undermines the persuasiveness of the findings; additionally, the authors could strengthen their work by incorporating comparisons with *rubric-based* approaches or methods that leverage stronger models for evaluation.

The remaining points will be addressed in the **Questions** section.

**Questions:**

1. Why should "more confident responses contribute more significantly to the final decision"? This claim requires substantiation through either theoretical derivation or empirical evidence;

2. Why can **std of topk_diff** effectively characterize **confidence**? Tokens representing logical relationships inherently exhibit large topk-diff values; if such tokens appear more frequently, the resulting lower confidence seems counterintuitive and lacks theoretical justification;

3. Why is "a consensus derived from high-confidence responses more reliable than one based on diverse low-confidence outputs"? This assertion necessitates validation through theoretical analysis or experimental results;

4. Why is "decisiveness more valuable during moments of high uncertainty"? Please provide theoretical derivation or empirical evidence to support this claim;

5. Why are comparisons with *rubric-based* approaches or methods leveraging stronger models for judgment absent? These alternatives should also be viable solutions to the RLIF problem;

6. The ablation studies require more in-depth analysis. For instance, why does **Credibility-Calibrated Pseudo-Labels** yield almost no positive effect? Why does **Decisive-Path-Reward** contribute minimally to performance improvement?

---

### Official Review · Reviewer_cW87 · 2025-10-31

**Soundness:** 3
**Presentation:** 2
**Contribution:** 2
**Rating:** 4
**Confidence:** 4

**Summary:**

This work aims to solve two key problems in existing work, Test-Time Reinforcement Learning (TTRL), a setting for LLM self-evolution on unlabeled test data : 1) the fragility of pseudo-labels and sparse rewards from majority voting , and 2) the neglect of reasoning process quality. It introduces COMPASS, a composite reward mechanism that combines a "Dual-Calibration Answer Reward" (DCAR) for more robust outcome scoring and a "Decisive Path Reward" (DPR) to reward the reasoning journey itself. Experiments demonstrate that this approach generally improves performance over the TTRL baseline across several reasoning tasks.

**Strengths:**

1. The work addresses the highly significant and challenging problem of LLM self-evolution using unlabeled data in a test-time setting.

2. The proposed method is a direct and effective improvement over the TTRL baseline, thoughtfully addressing its key weaknesses in reward sparsity and process-agnosticism .

**Weaknesses:**

1. Fragility of the DPR Heuristic: The process reward (DPR) is shown to be brittle; it fails and degrades performance on the LLaMA3.2-1B model, as the paper admits its guiding heuristic (rewarding decisiveness in high-entropy states) reinforces "fundamental confusion" in less capable models .

2. Extensive Reward Engineering: The method's success appears to rest on a complex combination of specific, fine-tuned heuristics (e.g., the exact formulas for confidence and credibility), which are not well-justified over simpler alternatives (see Question 1).

3. Narrow Scope and Limited Comparison: The contribution feels more like an incremental "patch" for TTRL rather than a general method. The experiments narrowly compare only against this single baseline, failing to benchmark against other RLIF approaches (like RENT or EMPO) that also tackle self-rewarding.

Overall, I am at a borderline. My main concern is whether the contribution is broad enough to be more than a "patch for TTRL." Given the potential, I would consider raising my score after the rebuttal.

**Questions:**

1. Confidence Metric Justification: Can you provide more details from the "empirical analysis" mentioned Line 220, that led to defining confidence as exp(standard deviation of topk diff)? Why was this specific metric chosen over simpler, more common ones like average sequence log-likelihood or the mean topk_diff? How sensitive is performance to this exact formulation?

2. DPR vs. Entropy Minimization: Your DPR uses high entropy as a positive weight to reward decisiveness. This philosophy seems to directly contradict other RLIF work like RENT, which aims to minimize entropy (Line 134). Could you comment on this apparent contradiction and justify why your approach is superior to simply rewarding low-entropy (high-certainty) outputs overall?

3. Computational Cost: The paper claims "relatively low computational costs" (Line 340); compared to what? It appears that this method will cost more compared to other test-time methods. Could you please quantify this additional cost relative to the TTRL baseline?

---

### Official Review · Reviewer_nvai · 2025-11-01

**Soundness:** 2
**Presentation:** 2
**Contribution:** 2
**Rating:** 2
**Confidence:** 4

**Summary:**

This work proposes COMPASS, a method for training LLMs using RL on unlabeled test data. COMPASS introduces (1) Dual-Calibration Answer Reward (DCAR) to calibrate majority voting with confidence measures, and (2) Decisive Path Reward (DPR) to optimize the quality of the thought process. Specifically, DCAR uses a confidence-calibrated self-consistency score to obtain the pseudo-label, and further propose a metric to assess the credibility of this pseudo-label. DPR is specifically designed to evaluate the model’s reasoning pathway, encouraging decisive actions at critical junctures.

The authors claim DCAR enables the model to prioritize learning from high-credibility consensus, while DPR provides a direct and dense supervisory signal for optimizing the reasoning path.

**Strengths:**

1. The research on RL without explicit labels for reasoning is a promising direction.

2. This paper attempts to address the fragility of pseudo-labels and evaluates the process quality, which is a good idea.

3. The design of DPR is very interesting, but I believe it may lack some empirical evidence to support it.

4. The results outperform baseline TTRL.

**Weaknesses:**

I find the claim/method of this paper is not very convincing, and the evaluation is limited. I discuss relevant weaknesses below.
### Method
The biggest problem is that the author makes many hypothetical claims without supporting empirical evidence or literature, which makes the argument unconvincing.

1. In line 218 and 241, the authors claim that "*We hypothesize that more confident responses should contribute more significantly to the final decision*" and "*Our underlying hypothesis is that a consensus derived from high-confidence responses is more reliable than one based on diverse low-confidence outputs*".
But it's not clear whether this hypothesis applies to both simple and difficult questions. Moreover, some studies [1][2] figure out that LLMs may "make mistakes with confidence". I suggest providing evidence to support those hypotheses.

2. In line 298, the authors claim that "Our central hypothesis is that decisiveness is more valuable during moments of high uncertainty". This also lacks some empirical evidence, and it's unclear whether it's related to the difficulty of questions. The author argues that tokens with higher certainty are better at this point and should be assigned a higher score. But will this impair the model's exploratory ability in RL and encourage overconfidence, which causes faster entropy collapse?

### Experiments
1. Concerns on baseline comparison. Only one baseline TTRL was compared. I suggest adding the comparison with supervised baselines to see the effectiveness of COMPASS, such as GRPO with ground truth labels.

2. Concerns on evaluation benchmarks. The evaluation benchmarks are limited on math and physics, I suggest adding the results on general reasoning domains like MMLU-Pro and StrategyQA to see the effectiveness of the proposed method.

2. Concerns on the experimental setting. Although this paper follows the pattern of Test Time Reinforcement Learning proposed by TTRL, I don't think it's reasonable. **Evaluating on the training set is hard for me to understand.** Recent studies reveal that there may be potential data contamination in the Qwen model of popular benchmarks like MATH and AMC [7]. Consequently, training and testing on the same contaminated benchmarks on Qwen2.5 series may be unreliable.

3. Concerns on evaluation results. I notice some evaluation results are inconsistent with those provided by the official Qwen and Llama. For example, Qwen2.5-7B achieves 36.4 accuracy on GPQA [8], which is even better than the performance of COMPASS reported in this paper. Besides, the official performance of Llama3.2-1B-Instruct on MATH and GPQA is 30.6 and 27.2 respectively [9], which seems inconsistent with the reported 24.7 and 23.8 in the paper.

### Presentation
The coordinate axes and fonts in Figure 2, 3 are small, which may affect the reading experience.

### Missing References
I suggest citing relevant works that also explore unsupervised reinforcement learning [3][4][5][6].

---

[1] Mind the Confidence Gap: Overconfidence, Calibration, and Distractor Effects in Large Language Models. arXiv preprint:2502.11028

[2] Why Language Models Hallucinate. arXiv preprint:2509.04664

[3] Learning to Reason without External Rewards. arXiv preprint:2505.19590

[4] Consistent Paths Lead to Truth: Self-Rewarding Reinforcement Learning for LLM Reasoning. arXiv preprint:2506.08745

[5] Reinforcing General Reasoning without Verifiers. arXiv preprint:2505.21493

[6] Can Large Reasoning Models Self-Train? arXiv preprint:2505.21444

[7] Reasoning or Memorization? Unreliable Results of Reinforcement Learning Due to Data Contamination. arXiv preprint arXiv:2507.10532

[8] Qwen2.5 Technical Report. arXiv preprint:2412.15115

[9] https://huggingface.co/meta-llama/Llama-3.2-1B-Instruct

**Questions:**

1. In Line 219, the authors claim "Through empirical analysis, we found that the standard deviation of topk_diff across a generation trajectory correlates strongly with the correctness of the final answer". I want to see the empirical analysis and evidence.

2. In Eq. (3), how is $std_t$ calculated?

---

### Official Review · Reviewer_7VGZ · 2025-11-01

**Soundness:** 2
**Presentation:** 2
**Contribution:** 2
**Rating:** 2
**Confidence:** 3

**Summary:**

- The paper proposes COMPASS, a test-time reinforcement learning method that lets large language models self-improve on unlabeled data using intrinsic rewards.
- It introduces two reward components: Dual-Calibration Answer Reward (DCAR) for more reliable pseudo-labels and Decisive Path Reward (DPR) for rewarding confident, high-quality reasoning steps.
- Experiments show that COMPASS consistently outperforms prior test-time RL methods, e.g., TTRL, across multiple reasoning benchmarks and model sizes.

**Strengths:**

- COMPASS effectively enables self-improvement without labeled data, making reinforcement learning scalable to unlabeled reasoning tasks.
- The proposed DCAR and DPR components are well-motivated and complementary, jointly addressing reward reliability and reasoning quality.

**Weaknesses:**

- The method still depends on self-consistency assumptions, which can reinforce systematic model biases or errors.
- The paper should include more challenging and diverse benchmarks to better demonstrate the generality of COMPASS.
- Error bars or statistical significance tests are missing, making it hard to assess the reliability of reported improvements.
- The performance gains over TTRL are relatively modest, raising questions about the practical impact of the proposed method.
- The paper provides limited analysis of failure cases, which could clarify when and why the method underperforms.

**Questions:**

- Can the authors clarify how the confidence and credibility metrics in DCAR interact when they disagree (for example, high confidence but low credibility cases)?
- Do the intrinsic rewards correlate with ground-truth correctness when labels are available?
- How does the method perform if the base model's confidence calibration is poor or systematically biased?
- Could the authors report the variance or standard deviation across multiple runs to assess the reliability of the reported improvements?
- What happens if the model generates reasoning paths of varying lengths?
- In Figure 4, the performance curves of different methods appear very similar, suggesting that the claimed improvements may be marginal or not clearly demonstrated in this comparison.
- In Line 16, consentrates -> concentrates
- In Line 18, absense -> absence
- In Line 24, desiciveness -> decisiveness
- In Line 215, algorithm -> Algorithm
- In Line 324, Specifically,we -> Specifically,(SPACE)we
- Equation (9) is not presented in a standard mathematical form; the use of a large opening parenthesis makes the expression unclear.
- In the caption of Figure 3, comparision -> comparison
- The reference formatting and citation usage need improvement; the authors should update the BibTeX entries for accuracy and consistently use \citep and \citet.
- Figures should use larger font sizes to improve readability.

**Details Of Ethics Concerns:**

There is no particular ethical concern.

---

### Note · Authors · 2025-11-17

I have read and agree with the venue's withdrawal policy on behalf of myself and my co-authors.